# Wildlife Management Practices Associated with Pathogen Exposure in Non-Native Wild Pigs in Florida, U.S.

**DOI:** 10.3390/v11010014

**Published:** 2018-12-26

**Authors:** Amanda N. Carr, Michael P. Milleson, Felipe A. Hernández, Hunter R. Merrill, Michael L. Avery, Samantha M. Wisely

**Affiliations:** 1Department of Wildlife Ecology and Conservation, University of Florida, Gainesville, FL 32611, USA; carrmandie@gmail.com (A.N.C.); fhernandeu@uc.cl (F.A.H.); 2Present Address: Biology Department, Western Washington University, Bellingham, WA 98225, USA; 3United States Department of Agriculture, Animal and Plant Health Inspection Service, Wildlife Services, Gainesville, FL 32641, USA; michael.p.milleson@aphis.usda.gov (M.P.M.); gibs8387@gmail.com (M.L.A.); 4School of Natural Resources and Environment, University of Florida, Gainesville, FL 32601, USA; 5Department of Agricultural and Biological Engineering, University of Florida, Gainesville, FL 32603, USA; hmerrill@ufl.edu

**Keywords:** Aujeszky’s disease, *Brucella* spp., brucellosis, harvest-disease dynamics, landscape epidemiology, pathogen emergence, pseudorabies virus, wildlife disease management, wild pigs

## Abstract

Land use influences disease emergence by changing the ecological dynamics of humans, wildlife, domestic animals, and pathogens. This is a central tenet of One Health, and one that is gaining momentum in wildlife management decision-making in the United States. Using almost 2000 serological samples collected from non-native wild pigs (*Sus scrofa*) throughout Florida (U.S.), we compared the prevalence and exposure risk of two directly transmitted pathogens, pseudorabies virus (PrV) and *Brucella* spp., to test the hypothesis that disease emergence would be positively correlated with one of the most basic wildlife management operations: Hunting. The seroprevalence of PrV-*Brucella* spp. coinfection or PrV alone was higher for wild pigs in land management areas that allowed hunting with dogs than in areas that culled animals using other harvest methods. This pattern did not hold for *Brucella* alone. The likelihood of exposure to PrV, but not *Brucella* spp., was also significantly higher among wild pigs at hunted sites than at sites where animals were culled. By failing to consider the impact of dog hunting on the emergence of non-native pathogens, current animal management practices have the potential to affect public health, the commercial livestock industry, and wildlife conservation.

## 1. Introduction

The One Health approach to studying disease acknowledges the interdependent relationship between humans, animals, and the environment that they share [1]. One Health perspectives further acknowledge, either implicitly or explicitly, that the ways in which landscapes are used and managed can affect the emergence and expansion of zoonotic and wildlife diseases [2,3,4,5]. While the link between changes to the physical landscape and outbreaks of infectious diseases is well-established [6,7,8,9], empirical evidence of the relationship between pathogen emergence and land use policy and management is lacking.

Hunting is a land use that has been managed in the United States since the early 20th Century. In the last several decades, hunting has been proposed as a means of wildlife disease control. This assertion assumed that hunting would truncate age structure and reduce population density, thus removing older, diseased individuals from the population and reducing density-dependent disease transmission [10,11,12,13,14,15,16]. Alternatively, it has been proposed that population-level responses to hunting, such as elevated birth rates or juvenile survival due to density-dependent host population growth, may instead increase the supply of disease-susceptible individuals and pathogen transmission [13,17,18,19]. In addition to inducing population-level responses to harvest, hunting in general [20] (and hunting with dogs in particular [21,22]) has been implicated in causing physiological changes associated with stress in hunted species. This relationship has important implications for the health of hunted wildlife populations, because chronic stress is associated with immunosuppression [23]. Despite the contested nature of the hunting-disease debate, empirical studies of the effect of harvests on disease dynamics are largely absent from the literature [24].

The potential for hunting to alter the disease dynamics of a system should be the greatest for non-native invasive species with large pathogen loads. Exotic invasive species are typically characterized by a high reproductive capacity that responds positively to medium and even high levels of harvest [25]. For example, wild pigs (*Sus scrofa*) are a ubiquitous exotic species in the U.S. that exhibit strong density-dependent demography and respond rapidly to harvest pressure by increasing fecundity [17,18,19,24]. Two non-native infectious pathogens of swine that are directly applicable to the study of harvest impacts on disease rates and transmission are pseudorabies (PrV, or Aujeszky’s disease), caused by the virus *Suid herpesvirus 1*, and brucellosis, caused by the bacteria *Brucella* spp. Direct swine–swine transmission of each pathogen typically leads to lifelong infection accompanied by neutralizing antibodies [26,27,28]. Though mortality is rarely associated with PrV and *Brucella* spp. in adult wild pigs [26,27,29], both pathogens can be lethal to fetal and juvenile pigs, wildlife, domestic animals, and, in the case of brucellosis, humans [30,31,32,33].

Florida is estimated to have >1 million free-ranging wild pigs, and they have been documented in all 67 counties [34]. Nonetheless, population size has not been estimated locally, nor have the number of harvested animals been enumerated. As a species with high ecological plasticity, they are found in a wide variety of habitat types and feed on a wide variety of plants and animals [35]. Both natural and human-assisted movement of pigs have contributed to their historical and recent expansion in Florida [36]. The management of wild pigs is complicated by their non-native status: On private lands, they are considered livestock and may be removed at any time and in any manner. On public lands that allow recreational hunting, they are the second-most popular game species (second only to white-tailed deer), and hunting methods vary from site to site. In areas that allow hunting, harvest quotas vary from none to some. In addition, on both private and public lands, wild pigs are frequently harvested for the purpose of animal control, are transported live to an abattoir for slaughter, or are transported live to privately owned game preserves for release and subsequent hunting [34]. Thus, in Florida, there are a wide variety of management actions taken [37], yet very little understanding of population and disease dynamics resulting from those actions. Considering the ubiquity of wild pigs in Florida and the economic and health threats posed by PrV and *Brucella* spp. to people, wildlife, and domestic animals, understanding the impacts of management actions on the emergence of these pathogens is important to informing future management decisions regarding wild pig populations.

In this study, we aspired to assess the effects of a singular harvest method (i.e., hunting with dogs) on the processes of disease dynamics and transmission. To this end, we compared seroprevalence and exposure risk for two pathogens among wild pig populations that were hunted with dogs versus populations that were harvested by other means. We chose to compare hunting with dogs to other harvest methods for two reasons: (1) Hunting with dogs is a popular method of harvest throughout the southeastern United States, and (2) hunting with dogs has been documented to increase the stress levels of both the targeted animal and the animals that are pursued but not killed [21,22]. Using almost 2000 serological samples collected from wild pigs at over 80 public and private sites across the geographic range of the state, we tested the hypothesis that the seroprevalence and risk of transmission of PrV and *Brucella* spp. would be increased by this form of harvest. Herein, we show that the prevalence of PrV exposure and PrV-*Brucella* spp. co-exposure, as well as individual risk of exposure to PrV, were greater in sites hunted with dogs than sites utilizing other harvest methods, but these differences were not found for *Brucella* spp. alone. Our results illustrate the necessity of including a One Health approach to understand the epidemiology of a disease and to incorporate effective wildlife management decisions to mitigate disease.

## 2. Materials and Methods

Data for this study included eight years (July 2007 to December 2014) of samples that were collected across Florida as part of a national wild pig disease monitoring effort led by the United States Department of Agriculture (USDA), Animal Plant and Health Inspection Service, Wildlife Services’ National Wildlife Disease Program (Figure 1). We collected 35 mL of blood from euthanized wild pigs on sites where animal control work was being conducted by the USDA, or from wild pigs that were killed in dog hunts at check stations where public hunts were managed by state wildlife or land management agencies. Regardless of the carcass acquisition method, samples were collected within 5 h of death. Using body size, reproductive characteristics, and dental structure as guidelines, we estimated age as one of three classes: Juvenile (<2 months), sub-adult (2 months–1 year), or adult (≥1 year) [38]. We included a sample in our analyses if the pig was unequivocally determined to be either seropositive or seronegative for PrV and *Brucella* spp.; the geographic location at which the sample was collected experienced either (1) dog hunting (*n* = 895) or (2) other forms of nonrecreational harvest for animal control (i.e., population management by live trapping or shooting without pursuit, *n* = 1096); the age of the animal was known; and the sex of the animal was known. We excluded sites with recreational hunting that only permitted hunting from tree stands or by stalking (*n* = 42). Total sample size after omissions was 1991 wild pigs (157 juveniles, 245 sub-adults, 1589 adults), with roughly equivalent sex ratios within the sample population (1000 females, 991 males). The relative proportion of pigs in each age class did not differ significantly (α = 0.05) between sites hunted with dogs and at sites with other forms of harvest (${Χ}_{2}^{2}=5.35,\;p=0.070$). Though there were slightly more females on sites with dog hunting (0.52 of all samples) than on sites with other forms of harvest (0.48 of all samples;${\;Χ}_{1}^{2}=3.94,\;p=0.047$), the difference was not biologically meaningful. No animals were euthanized for the purposes of this study. The University of Florida’s Institutional Animal Care and Use Committee approved this study (Protocol 201307937) on 8 May 2013.

Serological tests indicated the presence or absence of host antibodies to a pathogen. Because both PrV and *Brucella* spp. induce a lifelong infection and antibody production, a positive antibody test indicated that an animal was either exposed but not infected, or exposed and infected. Seroprevalence data for PrV and *Brucella* spp. have previously been used to determine relative pathogen prevalence and risk of transmission in wildlife populations (References [42,43], methodology reviewed in Reference [44]). Serological tests for PrV and *Brucella* spp. were performed at one of four designated National Animal Health Laboratory Network facilities in the United States: The University of Georgia Tifton Veterinary Diagnostic Laboratory, the Washington Animal Disease Diagnostic Laboratory, the Bronson Animal Disease Diagnostic Laboratory, or the Wisconsin Veterinary Diagnostic Laboratory. From July 2007 to September 2010, serum samples were screened for PrV antibodies using automated latex agglutination (LAT), and from October 2010 to December 2014, serum samples were screened using the PrV-gB enzyme-linked immunosorbent assay (ELISA). To screen for *Brucella* spp., an acidified antigen (card) test was used intermittently from December 2007 to September 2011; a rivanol (RIV) test was used for samples between May 2008 and February 2009; and a fluorescence polarization assay (FPA) was used extensively from October 2010 to December 2014.

To account for hierarchical uncertainty in diagnostic sensitivity, we estimated the probability of antibody detection at the state, site, and assay level in order to achieve a more accurate estimate of serological prevalence [45]. Assuming the presence (not prevalence) of disease was static at each site through time (e.g., References [27,46]), we used a year-to-year capture (i.e., detection) history approach within the software PRESENCE version 12.12 [47] to estimate the probability of antibody detection at the state and site levels. Using separate site-level detection probabilities for hunted and culled populations, we also calculated corrected probabilities of occurrence of PrV and *Brucella* spp. at each site. Because each animal was only sampled once, we could not estimate the probability of antibody detection for a given individual. We estimated the probability of antibody detection for each serological test by averaging the sensitivities presented in the existing literature (e.g., References [48,49,50]). For the *Brucella* spp. assays, we averaged the sensitivities for *Brucella abortus* due to the absence of studies evaluating the detection of *Brucella* spp.-specific antibodies. Compounded probabilities of detection were calculated as the product of probabilities at all levels (state, site, and assay). Because the detection of antibodies was approximately equivalent across treatments (see “Results” for details), all subsequent analyses used the apparent, uncorrected values of serological incidence and prevalence.

We tested our hypotheses regarding apparent seroprevalence (i.e., the proportion of wild pigs exposed to a given pathogen) and individual exposure risk separately. To first compare seroprevalence in sites with dog hunting to sites with other forms of harvest, we performed a contingency table Chi-squared analysis comparing the expected and observed frequencies of pigs exposed to no pathogen, PrV only, *Brucella* spp. only, and both pathogens. Expected frequencies were calculated by multiplying the number of pigs in a given exposure category by the proportion of pigs in each harvest category (i.e., 0.45 from hunted populations, 0.55 from populations with other forms of harvest). We then used multi-model inference within an information theoretic framework [51] to determine the magnitude of the effect and relative importance of dog hunting (i.e., hunting = 1 and other harvest = 0), age (i.e., juvenile, sub-adult, or adult), and the interaction between dog hunting and age on the risk of PrV and *Brucella* spp. exposure, independently. The response variable was the serostatus of an individual pig, coded as 0 for no antibodies detected (no apparent exposure) and 1 for antibodies detected (apparent exposure) for each pathogen. All models included a random effect of site to account for stochastic spatial variation in disease prevalence, and were fit using a binomial error structure with a logit link. We included age as a predictor variable because previous studies have indicated that the seroprevalence of both PrV and *Brucella* spp. increases with age in wild pigs (e.g., References [27,51,52,53]). Because most evidence suggests sex has a limited impact on either PrV or *Brucella* spp. seroprevalence in wild pigs [27,52,53,54], we excluded sex from our analyses. Though we would have liked to have included population density as a covariate because of its strong association with disease prevalence [42,55,56], no local or regional estimates of wild pig density exist within the state of Florida.

Because we could not rule out any combination of variables, our model set for each pathogen included the factorial combinations of hunting status of the site where the sample was collected, age of the animal, and the interaction between hunting status and age (model set *n* = 4). All models were fit using the R [41] package lme4 version 1.1-13 [57]. We used the R package MuMIn version 1.40.0 [58] to rank the models by the Akaike information criterion (AIC) and to calculate the model-averaged coefficient and relative importance of each term [51]. Coefficients were averaged over all models and weighted by model likelihood. Since we used logistic regression models, we also calculated odds ratios as the exponentiated coefficients. The relative importance of each variable was determined as the sum of the Akaike weights of the models in which they appeared [51]. For each term, we used α = 0.05 for the determination of statistical significance, and α = 0.10 as the threshold for borderline significance.

## 3. Results

### 3.1. Probability of Disease Detection

The analysis of hierarchical uncertainty in serological results provided insight into the sensitivity and bias in our samples (Table 1). Wild pigs tested seropositive for PrV and *Brucella* spp. in Florida every year, suggesting that the probability of antibody detection (*p*) at the state level was 1.00. The year-to-year probability of PrV antibody detection was *p* = 0.96 for sites with dog hunting and *p* = 0.72 for sites with other forms of harvest (overall *p* = 0.81, Appendix A). For *Brucella* spp., site-level antibody detection was *p* = 0.73 for hunted sites and *p* = 0.56 for sites with other forms of harvest (overall *p* = 0.63, Appendix A). Despite the variation in detection probabilities between dog-hunted and other harvest sites, the corrected probabilities of occurrence were approximately 20% higher for PrV and 30% higher for *Brucella* spp. in sites with dog hunting compared to sites with other forms of harvest (Appendix A).

For PrV assays, comparative studies have found that by the second week of exposure, both LAT and ELISA have near 100% sensitivity (Table 1). The probability of antibody detection varied more strongly among *Brucella* spp. serological tests, with FPA exhibiting the highest levels of sensitivity (Table 1). Given the general superiority of FPA, we used the FPA results for analysis in the rare cases where more than one test was implemented for the same sample. The compounded detection probabilities suggested that we identified 8 out of 10 PrV antibody-positive individuals, regardless of serological test. We detected approximately 1 out of every 3 *Brucella* spp. antibody-positive wild pigs screened by RIV, 1 out of every 2 screened by card, and 2 out of every 3 screened by FPA.

### 3.2. Effect of Dog Hunting on the Prevalence of Disease

The contingency table analysis indicated that seroprevalence differed significantly between wild pigs that were hunted with dogs versus those that were harvested without dogs (${Χ}_{3}^{2}=59.33,\;p<0.001$; Table 2). The frequency of pigs with zero pathogen exposure was 15% lower than expected at sites with dog hunting, and 12% higher than expected at sites with other forms of harvesting. Conversely, the frequency of exposure to PrV was 18% higher than expected for dog-hunted populations, and 14% lower than expected for populations with other forms of harvesting. Frequency of exposure to both PrV and *Brucella* spp. was 40% higher than expected at sites with dog hunting, and 34% lower than expected at sites with other forms of harvest. Observed exposure to *Brucella* spp. alone did not deviate from expected frequency for either treatment.

### 3.3. Effect of Hunting on the Odds of Pathogen Exposure

The model that best fit the data for PrV exposure included the effects of harvest method, age, and the interaction between harvest method and age (Table 3). These three variables explained 20% of the variation in the data, and the random site effect explained another 30% of the variation. Independently, harvest method and age were equally important in predicting PrV exposure (Table 4). Dog hunting caused a roughly twelvefold increase in the odds of exposure to PrV (95% CI = 7.01, 16.61) compared to other harvest methods. Adult pigs were ten times more likely to be exposed to PrV than juveniles (95% CI = 6.31, 13.71), but the odds of exposure did not differ between juveniles and sub-adults (95% CI = −1.56, 5.76). The effect of age did not vary significantly by harvest treatment.

The best-fitting model for *Brucella* spp. exposure included only the main effects of harvest method and age. These main effects explained 13% of the variation in the data, and the random site effect explained an additional 32% of the variation (Table 3). In contrast to PrV, age was more important than harvest method for predicting the seroprevalence of *Brucella* spp. (Table 4). The harvest method did not significantly affect exposure to *Brucella* spp. (95% CI = −1.34, 7.53), but adults were about 12 times more likely to be exposed to *Brucella* spp. than juveniles (95% CI = 7.46, 16.15). Risk of exposure was also about five times higher for sub-adults than for juveniles, but the effect was only borderline significant (95% CI = −0.04, 9.15). As with PrV, the effect of age did not vary significantly between dog-hunted populations and populations harvested by other means.

## 4. Discussion

Our study detected higher disease exposure in populations that were hunted by dogs than in those that were harvested by other methods, regardless of pig age. Contrary to classical theories identifying hunting as a means of wildlife disease control [10,11,12,13,14,15,16], these findings support the predictions of a more recent harvest-disease dynamics model, which proposed that hunting increases the prevalence of disease in species with a high reproductive output, such as wild pigs [24]. This positive association between dog hunting and disease exposure could be the result of several phenomena or the interaction between them: (1) Density-dependent demography [13,24], (2) immunosuppression due to elevated stress levels [20,21,22], and/or (3) increases in contact rates due to the disruption of animal movement and social structure [64,65]. Though this observational study was statistically robust, we did not have the experimental data necessary to conclude that dog hunting *caused* heightened disease exposure: However, these results identified a need to consider the elevated prevalence of disease as a side effect of this particular harvest practice.

The higher prevalence of PrV and PrV-*Brucella* spp. co-exposure among wild pigs in sites hunted with dogs compared to other harvest methods is not well explained by density-dependent responses to harvest or by systematic differences in population densities. Population-level responses to animal removal, either by recreational hunting or by animal control, could induce elevated birth rates in both scenarios, which increases the supply of disease-susceptible individuals [13,24]. A previous work on European wild boar proposed that these demographic responses to hunting allow the persistence of diseases such as classical swine fever in free-ranging wild boar populations [17,18] and that removal of up to 90% of the adult population is required per annum to keep populations from increasing in density [66]. However, because all sites had some level of removal, we could not attribute the apparent effect of hunting with dogs observed here to density-dependent demography. In addition, it seems unlikely that there were systematic differences in population densities between sites that allowed hunting with dogs and other sites that harvested pigs. Dog hunting in Florida is not preferentially prescribed for areas with higher densities of pigs. At sites that allow dog hunting, it is primarily a form of recreation, with population control as a secondary objective. Indeed, most land and wildlife managers employ baited trapping as preferred forms of targeted population control, particularly in areas with high densities of pigs that create conflicts with land use mandates such as nonhunting outdoor recreation (e.g., county or state parks) or highly endangered habitats (e.g., national wildlife refuges, state wildlife refuges, national parks).

Nondemographic factors, such as stress, may have contributed to the harvest-induced disease response for PrV, an alpha herpesvirus. Previous studies have linked harvest activities to elevated levels of stress in other wildlife populations [20,21,22], and elevated stress has been linked to immunosuppression [23]. Elevated stress has been shown to induce recrudescence of latent herpesvirus infection and intensify viral shedding in cattle with bovine herpesviruses [67] and mice with herpes simplex virus [68], and stress has been the suggested cause of Macacine alphaherpesvirus 1 recrudescence in rhesus macaques [69] and PrV recrudescence in swine [70,71]. Recrudescence of latently infected wild pigs would increase the risk of exposure and transmission via multiple routes of contact [31,70,72,73,74,75]. Though *Brucella* spp. may respond to general immunosuppression associated with chronic stress, it does not exhibit the stress-induced recrudescence of PrV, which may explain why we found no difference in *Brucella* spp. exposure between populations hunted with dogs and populations that were culled by other harvest methods. We did, however, find a positive response to harvest method in co-exposed animals. If PrV infection occurs during periods of low immune response, it would leave swine more vulnerable to secondary infections such as *Brucella* spp. [48]. This additional decline in immunity associated with PrV infection may explain why dog-hunted pigs were more likely to be co-exposed to both PrV and *Brucella* spp., but not more likely to be exposed to *Brucella* spp. alone. Testing this hypothesis explicitly would require measuring stress hormone levels concurrently with immune function markers and pathogen exposure data in populations with and without dog hunting.

In addition to elevating stress and birth rates, hunting with dogs may increase disease transmission by altering animal movement and social structure, which in turn may increase contact rates and transmission among individuals. For example, multiple groups of wild pigs may concentrate in areas inaccessible to dogs during the hunting season. Indeed, a study of wild pig movement in Alabama, U.S., found that during the hunting season, wild pigs had a significantly smaller home range and exhibited differential preference in habitat use [64]. However, other studies have found no such effect [76]. Such concentration of typically disparate social groups could temporarily increase local pig densities and associated contact rates, facilitating disease transmission [65]. Future research tracking swine movement and contact rates during and after the hunting season would elucidate the role of movement and social structure in the harvest-disease dynamic in feral swine observed here.

The possibility of confounding environmental variables was possible but not probable, as our sampling of both harvest treatments occurred across the geographic breadth of the state and primarily on public lands. As habitat generalists, wild pigs are found in virtually every habitat type in Florida, but tend to concentrate in areas with canopy cover [37] and in areas that offer supplemental sources of feed (e.g., agriculture [66,77,78]) or temporary refugia from hunting [64,79]. If habitat type were a confounding variable that influenced population density and therefore disease prevalence, then we would expect land cover associated with increased density (e.g., agriculture) to have been higher in dog-hunted sites. However, sites without dog hunts had significantly more agricultural cover within 6 km of the collection site (*p* < 0.001, Hernández et al. in prep. [80]), suggesting that this land cover type was not confounding the observed pattern. In addition, our inclusion of site as a random variable in our mixed effect model accounted for stochastic spatial variation in our samples. Indeed, the detection of a significant effect of hunting with dogs, despite the large variation among sites (Table 3), bolstered the support for the association between harvest method and disease dynamics.

The finding that hunting may modulate disease dynamics of PrV and *Brucella* spp. has agricultural, public health, and conservation implications. PrV infection is rapidly fatal in cattle and carnivores [33] and has been implicated in multiple deaths of the endangered Florida Panther (*Puma concolor coryi*) [81]. Although PrV is not considered to be a disease agent in humans, *Brucella* spp. is a major zoonotic pathogen in Florida and the U.S., producing nonspecific, flu-like symptoms in humans with the potential for serious, lifelong musculoskeletal, cardiovascular, genitourinary, and neurological complications if untreated [30,82]. In Florida, human infection with *Brucella* spp. is the second-highest reported zoonotic infection after *Borrelia burgdorferi*, the causative agent of Lyme disease [83]. Commercial livestock in the U.S. are considered free of both PrV and *Brucella* spp., yet both pathogens are widespread in wild pig populations throughout the country, which creates the potential for reinfection of commercial herds [26,27,56].

By using a One Health approach to consider land use effects in the potential for transmission of pathogens that affect humans and animals, we are better able to make management recommendations about the control of disease in wild pigs. Considering the growing accumulation of evidence against hunting as an effective population control method in density-dependent species [24,84,85,86,87], a shift away from public hunts for the purposes of reducing population size warrants strong consideration. Targeted trapping, sharpshooting, and/or aerial gunning have been shown to reduce a larger proportion of the population, which is necessary to overcome density-dependent increases in reproduction, but they require greater resources to implement [83,88,89,90]. For other dog-hunted species, such as white-tailed deer (*Odocoileus virginianus*) and raccoons (*Procyon lotor*), managers should consider the potential to increase the risk of disease when making management decisions. Sustainable wildlife management generally assumes that harvest and infection rates are independent of one another [24]. However, recent models [24] and the results of this study suggest significant covariance between disease risk and hunting method. Harvesting policies that do not account for hunting-induced increases in disease prevalence and associated mortality may underestimate overall mortality rate and the probability of local extirpation of harvested populations.

Our One Health approach extends beyond the management of this invasive species and into conservation and public health policy. By identifying a land use (i.e., legal hunting with dogs) associated with higher disease prevalence and exposure risk, we have also identified areas with increased cross-species transmission potential. PrV is the third leading cause of mortality in Florida panthers (Mark Cunningham, pers. comm.). This population is expanding northward into the Kissimmee River Valley of Florida, where multiple publicly managed lands allow dog hunts. This vulnerable expanding population may benefit from a change in land use management policy to remove hunting with dogs as a hunting option in these areas. These areas are also hotspots for coinfection with *Brucella* spp., and thus wild pig hunters have a higher likelihood of exposure to this zoonotic pathogen. Increased hunter education about carcass handling and meat processing would be warranted in these public lands that allow dog hunts.

## Figures and Tables

**Figure 1 viruses-11-00014-f001:**
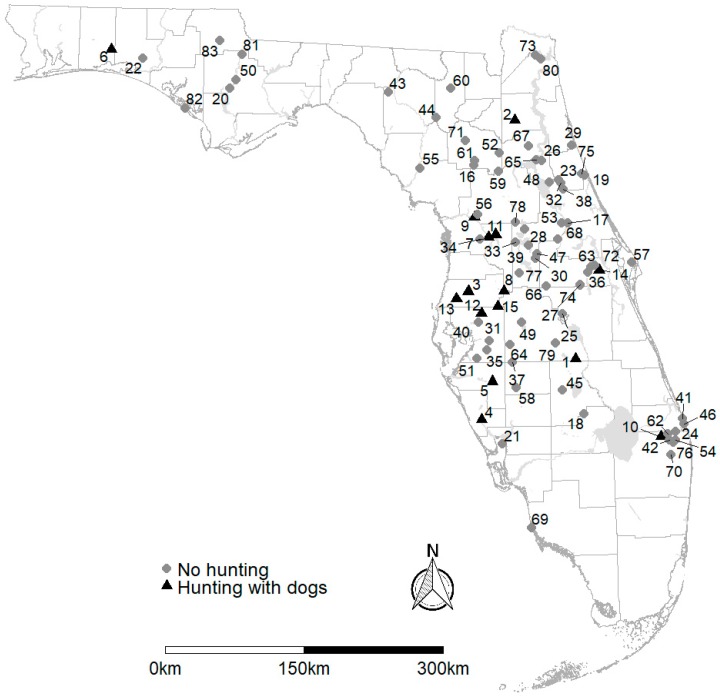
Sampling sites used to evaluate the exposure to pseudorabies (PrV) and *Brucella* spp. in wild pigs (*Sus scrofa*) in Florida, U.S. (2007–2014). Details (pathogen detection history, sample sizes, and apparent seroprevalence) for each site are provided in Appendix A (PrV) and Appendix A (*Brucella* spp.). County boundaries [39] and major water bodies [40] were downloaded from https://www.fgdl.org with no access or use constraints. The map was prepared using R version 3.4.0 [41].

**Table 1 viruses-11-00014-t001:** Hierarchical uncertainty for the detection probability (*p*) of pseudorabies virus (PrV) and *Brucella* spp. antibodies in wild pig populations of Florida, U.S. (2007–2014). LAT = latex agglutination test; ELISA = enzyme-linked immunosorbent assay; card = acidified antigen test; RIV = rivanol test; FPA = fluorescence polarization assay.

Pathogen	Level	*p*	Reference
PrV	State	1.00	This study
	Site	0.81	This study
	Assay		
	LAT	0.99	[59,60,61]
	ELISA	1.00	[59,60,61]
	Compounded		
	LAT	0.80	This study
	ELISA	0.81	This study
*Brucella* spp.	State	1.00	This study
	Site	0.63	This study
	Assay		
	Card	0.73	[62,63]
	RIV	0.60	[62,63]
	FPA	0.99	[63]
	Compounded		
	Card	0.46	This study
	RIV	0.38	This study
	FPA	0.62	This study

**Table 2 viruses-11-00014-t002:** Contingency table results for observed frequencies of antibody detection in four categories: No antibodies, pseudorabies virus (PrV) antibodies only, *Brucella* spp. antibodies only, and the occurrence of both antibodies in populations that were hunted with dogs and populations that were harvested by other means. Direction of arrow indicates whether observed frequencies (provided) were higher (↑) or lower (↓) than expected. Significance: ns = not significant; ~ = borderline significant (0.05 < *p* < 0.10); * *p* < 0.05; ** *p* < 0.01; *** *p* < 0.001.

Treatment	No Disease	PrV	*Brucella* spp.	PrV-*Brucella* spp.
Dog-hunted	426 *↓****	326 *↑***	42 ns	101 *↑****
Not dog-hunted	694 *↑****	291 *↓***	53 ns	58 *↓***

**Table 3 viruses-11-00014-t003:** Top three models for the response of pseudorabies virus (PrV) or *Brucella* spp. exposure to harvest method, wild pig age, and the interaction between harvest method and age.

Pathogen	Model	*K*	ΔAIC	*w_i_*	*R^2^_m_*	*R^2^_c_*
PrV	Harvest + Age + Harvest x Age	6	0.00	0.65	0.20	0.50
	Harvest + Age	5	1.24	0.35	0.18	0.48
	Age	4	12.55	0.00	0.06	0.46
*Brucella* spp.	Harvest + Age	5	0.00	0.77	0.13	0.45
	Harvest + Age + Harvest x Age	6	3.61	0.13	0.14	0.45
	Age	4	4.03	0.10	0.08	0.44

Notes: *K* is the total number of parameters (including intercept, random site effect, and residual error); ΔAIC is the difference in the Akaike information criteria of model *i* and the best-fitting model; *w_i_* is the Akaike weight of model *i*; *R^2^_m_* is the marginal *R*^2^ (i.e., the proportion of total variance explained by fixed factors only); and *R^2^_c_* is the conditional *R*^2^ (i.e., the proportion of total variance explained by both fixed and random factors).

**Table 4 viruses-11-00014-t004:** Model-averaged parameter estimates, odds ratios (ORs), and relative importance (RI) for the effects of harvest method, wild pig age, and the interaction between harvest method and age on pseudorabies virus (PrV) or *Brucella* spp. exposure. Values in bold denote statistical significance (*p* < 0.05) of the corresponding predictor variables. Italics signify borderline significance (0.05 < *p* < 0.10).

Pathogen	Variable	Coefficient	OR	SE_OR_	RI	*p*-value
PrV	(Intercept)	−3.62	0.03	1.93		< 0.001
	**Harvest**	**2.47**	**11.81**	**2.45**	**1.00**	**0.006**
	**Age**				**1.00**	
	**Adult**	**2.30**	**10.01**	**1.89**		**< 0.001**
	Sub-adult	0.74	2.10	1.87		0.234
	Harvest x Age				0.65	
	Harvest x Adult	−0.83	0.44	2.22		0.301
	Harvest x Sub-adult	−0.55	0.57	2.06		0.445
*Brucella* spp.	(Intercept)	−5.06	0.01	2.31		< 0.001
	Harvest	1.13	3.09	2.26	0.90	0.167
	**Age**				**1.00**	
	**Adult**	**2.47**	**11.80**	**2.22**		**0.002**
	*Sub-adult*	*1.52*	*4.56*	*2.34*		*0.075*
	Harvest x Age				0.13	
	Harvest x Adult	−0.08	0.92	1.77		0.884
	Harvest x Sub-adult	−0.04	0.96	1.78		0.942

Notes: Juvenile was the reference age class to which adult and sub-adult exposure odds were compared. SE_OR_ is the standard error of the odds ratio. Coefficients were averaged over all models, weighted by model likelihood. In models that did not contain a given variable, the coefficient of that variable was assumed to be zero.

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
