# Peer review of "Wildlife Management Practices Associated with Pathogen Exposure in Non-Native Wild Pigs in Florida, U.S."

_viruses, 2018, doi:10.3390/v11010014_

Round 1

Reviewer 1 Report

A nice and original article with a good collection of field data. My main suggestion:

Please include the formulae that explain how the antibody detection probabilities, corrected probabilities of occurrence, apparent seroprevalences, expected seroprevalences or compounded detection probabilities were calculated.

Other questions:

- the probability of antibody detection at the stae, site and assay level correspond to which column in Tables S1 and S2?

- It is not clear to me how many models and of which type were tested in total

- I do not understand the concept of harvest practices or harvest disease dynamic model- please explain what does it mean

- can infection in hunting dogs influence the status of any of the two infections studied in wild pigs?

- were the sites with no hunting that surrounded those with dog hunting influenced by the infection/exposure status of the dog hunting site nearby?

Author Response

Reviewer 1

Comments and Suggestions for Authors

A nice and original article with a good collection of field data. My main suggestion:

Please include the formulae that explain how the antibody detection probabilities, corrected probabilities of occurrence, apparent seroprevalences, expected seroprevalences or compounded detection probabilities were calculated.

We appreciate the need to make the methods transparent. A full mathematical accounting of occupancy modeling and defining the two parameters of interest P(occupancy) and P(detection) would require a lot of manuscript space. This is an accepted approach to defining prevalence in disease ecology and we provide multiple references that define its use. What we have done, however, is significantly increase the text that describes the procedure and its components (beginning line 157). We hope that this is sufficient for the reviewer.

Other questions:

- the probability of antibody detection at the stae, site and assay level correspond to which column in Tables S1 and S2?

Those tables correspond to site-level detection only, which is now explicitly stated in the captions.

- It is not clear to me how many models and of which type were tested in total

That information is now clarified. All models were mixed effect, logistic regressions (line 179), and there were four models total (lines 179-185).

- I do not understand the concept of harvest practices or harvest disease dynamic model- please explain what does it mean

You are correct, we were ambiguous with our language use. We have significantly revamped the terminology so that it is clear that we have hunting with dogs, or harvest by other means. Hunting implies recreational harvest, while harvest is the more generalized term that means removing animals from the population by any means and for any reason.

- can infection in hunting dogs influence the status of any of the two infections studied in wild pigs?

Infection is lethal to most non-suid animals, including hunting dogs (line 63). There is a rapid onset of disease in all carnivores, so the transmission of PrV or Brucella spp. to feral swine from hunting dogs is not a concern and has never been recorded.

- were the sites with no hunting that surrounded those with dog hunting influenced by the infection/exposure status of the dog hunting site nearby?

None of the sites were co-located and indeed many were far away from each other. Please refer to Figure 1. We do not feel that this was an issue. In addition, our inclusion of site as a random effects variable, accounts for the site to site variation in seroprevalence that is not associated with the treatment variable (harvest method).

Reviewer 2 Report

The paper attempts to determine whether hunting wild boar with dogs affects seroprevalence to pseudorabies virus and Brucella spp. The question is very relevant, given that increased hunting pressure is often proposed as a means to manage disease in wild population. Unfortunately, I feel the authors blur the line between association and causation and don’t provide adequate discussion on the limitations of their research, potential confounders and alternate explanations for their findings. This along with the apparent absence of table 2 in the manuscript, the inappropriate use of references (see below), leads me to the conclusion that this manuscript should not be published without major revisions.

A central question I had after reading this manuscript was if there were reasons why dogs were allowed to be used at some of these sites and not others. I would presume that the use of dogs may be related to habitat types and size, proximity to people and agriculture, high population densities of pigs and therefore significant damage resulting in increased use of dogs, etc. If so one or more of these factors may be acting as a confounder in this apparent relationship between hunting with dogs and higher seroprevalence. The reasons why some places allow hunting with dogs, and others do not, should be discussed and the possibility of confounders should be presented.

I was also disappointed in the author’s explanations of their findings. They speculate stress as the main explanation but make no mention of the effect hunting has on social structure, animal movement, etc. These too have the potential to affect disease dynamics.

Line 168: Reword  sentence “We included age to explain because….” Explain what?

Line 281: Elevated stress in wild pigs induces recrudescence of latent PrV infection and intensifies viral shedding [34-35]. Reference 34 is an experimental exposure with no apparent attempt at evaluating the effect of stress on viral shedding. In the discussion they speculate that stress of trapping and transportation may result in increased viral shedding but this in no way demonstrates that this is true. It is very disturbing to see speculation in other papers turned into statement of fact in subsequent papers. Reference 35 appears to be a case report and I suspect there is speculation about recrudescence as well. These two references do not support the author’s statement. I did not have time to investigate all other references but this example draws into question whether the use of references were appropriate in this manuscript.

Author Response

Reviewer 2

Comments and Suggestions for Authors

The paper attempts to determine whether hunting wild boar with dogs affects seroprevalence to pseudorabies virus and Brucella spp. The question is very relevant, given that increased hunting pressure is often proposed as a means to manage disease in wild population. Unfortunately, I feel the authors blur the line between association and causation and don’t provide adequate discussion on the limitations of their research, potential confounders and alternate explanations for their findings. This along with the apparent absence of table 2 in the manuscript, the inappropriate use of references (see below), leads me to the conclusion that this manuscript should not be published without major revisions.

We appreciate the reviewers concerns regarding the inferences we make, the caveats to our findings, and an inadequate background review of immune suppression and disease. WE have revamped the manuscript to address each of these concerns. Each of the above concerns is addressed in turn, below, and the accidental omission of Table 2 from the body of the text has been corrected.

A central question I had after reading this manuscript was if there were reasons why dogs were allowed to be used at some of these sites and not others. I would presume that the use of dogs may be related to habitat types and size, proximity to people and agriculture, high population densities of pigs and therefore significant damage resulting in increased use of dogs, etc. If so one or more of these factors may be acting as a confounder in this apparent relationship between hunting with dogs and higher seroprevalence. The reasons why some places allow hunting with dogs, and others do not, should be discussed and the possibility of confounders should be presented.

Dogs are used nearly exclusively for recreational hunting on private lands, state and federally regulated lands. The use of dogs is not universally allowed on state or federally owned lands but it is typically up to the land manager. It is however, a very popular form of hunting in the southern US. Dog hunts are not a management prescription for control or used in any particular habitat types, sizes, or densities of swine. Dog hunting is found throughout the state and throughout the southeastern United States. However, the reviewer is correct in that hunting in general is not allowed in periurban areas, although lethal animal control often is. A more thorough introduction to wild pig management is given in the introduction (starting line 66). These potentially confounding variables are now addressed in the discussion (beginning line 345).

I was also disappointed in the author’s explanations of their findings. They speculate stress as the main explanation but make no mention of the effect hunting has on social structure, animal movement, etc. These too have the potential to affect disease dynamics.

We have now clarified the evidence for stress and discussed alternative hypotheses (lines Paragraph beginning line 334).

Line 168: Reword  sentence “We included age to explain because….” Explain what?

Thank you for catching that word omission. The sentence has been modified.

Line 281: Elevated stress in wild pigs induces recrudescence of latent PrV infection and intensifies viral shedding [34-35]. Reference 34 is an experimental exposure with no apparent attempt at evaluating the effect of stress on viral shedding. In the discussion they speculate that stress of trapping and transportation may result in increased viral shedding but this in no way demonstrates that this is true. It is very disturbing to see speculation in other papers turned into statement of fact in subsequent papers. Reference 35 appears to be a case report and I suspect there is speculation about recrudescence as well. These two references do not support the author’s statement. I did not have time to investigate all other references but this example draws into question whether the use of references were appropriate in this manuscript.

As suggested by this reviewer we have included alternative hypotheses to stress induced recrudescence as the driver of the pattern we observed. When we do discuss it, we include a more extensive and well-established literature on the link between stress and herpesvirus recrudescence, yet make clear that in the references in question, it is simply speculation. Lines 318-323.

Round 2

Reviewer 2 Report

The manuscript has been significantly improved with these revisions. I did not check all of the references but I did detect some discrepancies. On line 427 they authors list reference 47 but I believe they mean 58. As well the link they provide for reference for reference 58 no longer works, although it may have worked in 2016 when they indicate they accessed it. As well, some references have been deleted, such as 53 in the bibliography but the numbering stays the same. The references need to be checked before publishing.